# An Efficient Digital Channelized Receiver for Low SNR and Wideband Chirp Signals Detection

Wenhai Cheng [1,2,3], Qunying Zhang [1,2,*], Wei Lu [1,2], Haiying Wang [1,2,3] and Xiaojun Liu [1,2]

1   Aerospace Information Research Institute, Chinese Academy of Sciences, Beijing 100190, China
2   Key Laboratory of Electromagnetic Radiation and Sensing Technology, Chinese Academy of Sciences, Beijing 100190, China
3   School of Electronic, Electrical and Communication Engineering, University of Chinese Academy of Sciences, Beijing 100190, China
*   Correspondence: qyzhang@aircas.ac.cn; Tel.: +86-010-5888-7408

**Abstract:** Synthetic aperture radar (SAR) is essential for obtaining intelligence in modern information warfare. Wideband chirp signals with a low signal-to-noise ratio (SNR) are widely used in SAR. Intercepting low-SNR wideband chirp signals is of great significance for anti-SAR reconnaissance. Digital channelization technology is an effective means to intercept wideband signals. The existing digital channelization methods have the following problems: the contradiction of reception blind zone and signal spectrum aliasing, high computational complexity, and low estimating accuracy for chirp signals with a low SNR. This paper proposes a non-critical sampling digital channelized receiver architecture to intercept chirp signals. The receiver architecture has no blind zone in channel division and no aliasing of signal spectrum in the channel, which can provide reliable instantaneous frequency measurements. An adaptive threshold generation algorithm is proposed to detect signals without prior information. In addition, an improved instantaneous frequency measurement (IFM) algorithm is proposed, improving low SNR chirp signals' frequency estimation accuracy. Moreover, a simple channel arbitration logic is proposed to complete the cross-channel combination of wideband signals. Simulations show that the proposed receiver architecture is reliable and robust for low SNR and wideband chirp signal detection. When the input SNR is 0 dB, the absolute frequency root-mean-square error (RMSE) of bandwidth and the center frequency is 0.57 MHz and 1.05 MHz, respectively. This frequency accuracy is great for radio frequency (RF) wideband systems.

**Keywords:** chirp signal detection; digital channelized receiver; instantaneous frequency measurement; channel arbitration



## 1. Introduction

Radar reconnaissance uses reconnaissance receivers to monitor the electromagnetic environment, intercept radiation source signals, and measure and analyze radiation source parameters [1]. Radar reconnaissance is an essential means of obtaining battlefield intelligence and a prerequisite for implementing electronic jamming and destruction [2]. Synthetic aperture radar (SAR) is capable of all-weather imaging reconnaissance and plays a vital role in high-tech warfare. Anti-SAR reconnaissance is an important research topic [3–5]. Wideband chirp signals with a low signal-to-noise ratio (SNR) are widely used in SAR [6–8]. Intercepting low SNR wideband chirp signals with high estimation accuracy is challenging. On the one hand, wide instantaneous bandwidths lead to high signal sampling rates, resulting in high processor speeds. However, the digital signal processing equipment's processing rate always lags behind the analog-to-digital converter's sampling rate (ADC) [9]. On the other hand, the low SNR will affect signal detection and increase the estimation error. In addition, computational efficiency should be taken into consideration due to the time-sensitive nature of radar reconnaissance. This paper aims to investigate wideband receiver architecture to intercept low SNR chirp signals efficiently.

Digital channelization techniques are a good solution for processor rate mismatch [10–12]. However, to have continuous coverage across the instantaneous bandwidth (IBW), the responses of adjacent channels are overlapped. For many traditional uniform channelization structures, such as polyphase filter structures [13,14] and FFT-based filter structures [15,16], overlapped channel response will cause signal spectrum aliasing due to the filter bandwidth being greater than the channel sampling rate, resulting in inaccurate spectrum measurement. A simple and efficient solution uses a non-critically sampled channelization structure [17–19]. By reducing the signal extraction multiple, the channel sampling rate is increased to ensure that the channel sampling rate is greater than the filter bandwidth [19]. However, existing non-critically sampled channelization structures cannot minimize the channel sampling rate because the channel number is multiple of the decimation number [17]. Another solution is to use a dynamic channelization structure [20–23]. Dynamic channelization adopts an analysis–synthesis structure. The original signal is channelized by the uniform filter bank first. Then, the output of the uniform filter bank is analyzed to judge the effective signals in channels. Finally, the effective signal is sent to the comprehensive filter bank to complete the reconstruction. The reconstructed wideband signal is used to measure the signal parameters The dynamic channelization method requires one more step of wideband signal reconstruction than the conventional method, which consumes many computing resources to meet the complete reconstruction conditions. They are not suitable for flexible and efficient radar countermeasure systems.

The channelization filter bank decomposes the wideband signal into multiple low-rate narrowband signals, requiring a frequency measurement module placed at the output of each channel due to the frequency estimation accuracy being limited to $\pm 1/2$ the channel bandwidth. The frequency measurement module is mainly implemented by the Coordinate Rotation Digital Computer (CORDIC) algorithm in the digital channelized receiver [24–26], which is simple and efficient. Nevertheless, the frequency measurement error cannot meet the requirement when the SNR is low. The advanced frequency estimation algorithms have great estimation accuracy [27–29], but they are unsuitable for the signal interception application due to the low calculation efficiency. Wideband signals are decomposed into narrowband signals through channelization filter banks, and the parameter measurement results of narrowband signals need to be combined to obtain wideband signal parameters. Traditional dynamic channelization algorithms can solve this problem by measuring parameters after signal reconstruction, but signal reconstruction and quadratic parameter estimation will consume computing resources [20]. The dynamic channelization algorithm based on sub-band spectrum detection does not need signal reconstruction [30,31], but the frequency estimation accuracy could be better due to different applications. In the field of digital channelization receivers, much literature focuses on a single digital channelization structure, and there are few studies on the combination of digital channelization structure and parameter measurement after channelization [17,24,32]. In [32], Zahirniak et al. proposed an efficient non-critical sampling channelized digital receiver structure and combined digital channelization technology with an instantaneous frequency measurement (IFM) algorithm. Based on the IFM results, the algorithm completes channel arbitration and determines whether the narrowband signals distributed in different channels belong to the same signal. Then, the narrowband signal parameters are merged according to the channel arbitration result. The algorithm has high computational efficiency and can adapt to wide instantaneous bandwidth. However, when the SNR of the input signal is lower than 5 dB, the estimation accuracy of the chirp signal parameters is low. In addition, the channelization sample rate cannot precisely match the prototype filter bandwidth and cannot be minimized because the channel number is multiple of the decimation number.

This paper proposes a digital channelized receiver architecture for wideband low SNR chirp signal detection. A non-critical sampling channelization structure based on weighted overlap-add (WOLA) is adopted, reducing the channel sample rate as much as possible under the condition that there is no blind zone and the signal spectrum is not aliased. In addition, an adaptive threshold generation algorithm is proposed to detect signals without

prior information. Moreover, an improved CORDIC based IFM algorithm is proposed, improving low SNR chirp signals' frequency estimation accuracy. The instantaneous frequency is used for channel arbitration, and a channel arbitration logic is proposed to complete the cross-channel combination of wideband signals. Simulation shows that the proposed algorithm is reliable and robust for low SNR and wideband chirp signal detection.

The rest of the paper is organized as follows. Section 2 presents the principle of channelization and the channel division method. Moreover, a digital channelized receiver architecture is also proposed. In addition, each part of the receiver architecture is introduced in detail. In Section 3, the reliability and robustness of the proposed receiver architecture are verified by simulation experiments, and the computational costs are analyzed. Finally, a conclusion is drawn in Section 4.

## 2. Methodology

This section presents the problem formulation and the proposed algorithm description for this study. The comparison table of symbols and its meanings used in this paper is shown in the Table 1.

**Table 1.** The comparison table of symbols and their meanings.

| Symbol | Meaning | Symbol | Meaning |
|---|---|---|---|
| $x(n)$ | Input sampling sequence | $f_s$ | System sampling rate |
| $h(n)$ | Prototype low-pass filter system function | $f_{ch}$ | Channel sampling rate after downsampling |
| $h_k(n)$ | The $k_{th}$ channel band-pass filter system function | $K$ | Channel number |
| $y_k(m)$ | The output sampling sequence of channel $k$ | $D$ | Downsampling decimation factor |
| $\omega_k$ | The center frequency of the $k_{th}$ channel | $\delta$ | Channel overlap factor for channel division |
| $\Delta\phi_k[m]$ | The phase difference between the $m$ and $m-1$ sampling point in channel $k$ | $\hat{\omega}_k^i$ | The estimated angular frequency of the $m$ sampling point in channel $k$ |

### 2.1. Problem Formulation

This section introduces the principle of channelization and the channel division method. The limitations of channelized receivers are explained.

#### 2.1.1. The Principle of Channelization

Wideband digital receivers need to consider a high sampling rate and a high processing rate that matches the sampling rate. However, digital signal processing devices' processing rate always lags behind the analog-to-digital converter's sampling rate. The channelization method uses a digital filter bank to convert the sampled high-speed data into a baseband signal and decimate it simultaneously, reducing the data rate so that the signal can be real-time analyzed and processed by the signal processor [10].

Under ideal conditions, the monitoring frequency band can be divided into $K$ channels. The filter banks for channelization is shown in Figure 1, where $h_k(n)(k = 0, 1, \cdots, K-1)$ is the band pass filter.

When the monitoring frequency band is evenly divided into $K$ channels, the bandwidth of the filter bank output signal $y_k(m)$ is $2\pi/K$. According to the Nyquist sampling theorem, the spectral structure of $y_k(m)$ will not change even though $y_k(m)$ is decimated by $K$ times. A simple implementation of uniform channelization for the complex signal is shown in Figure 2 [9], where the number of channels is $K$, the center frequency of each channel is $\omega_k(k = 0, 1, \cdots, K-1)$, and the decimation factor is $D, D \leq K$.

Ideally, the channel bandwidth is $2\pi/K$, and the frequency estimation accuracy is $\pm\pi/K$.

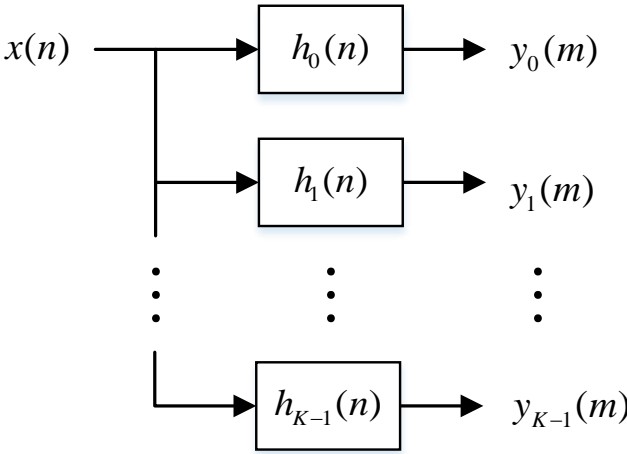

**Figure 1.** The filter banks for channelization.

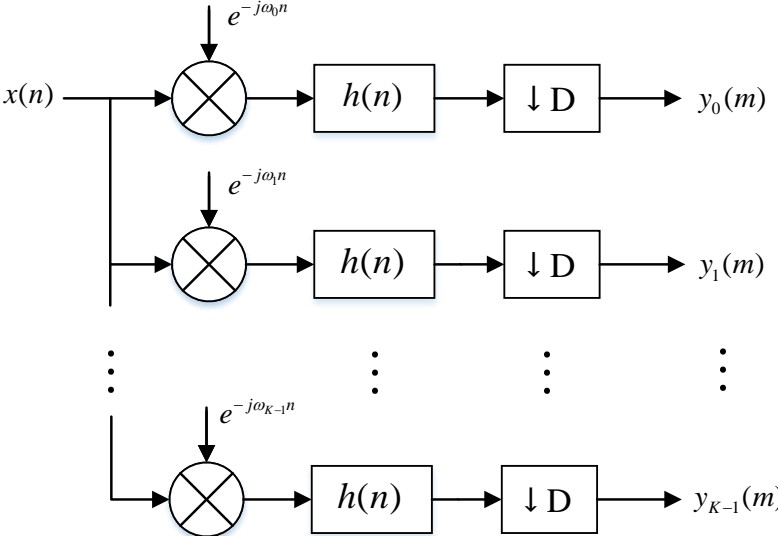

**Figure 2.** The complex signal down-conversion low-pass channelization structure.

### 2.1.2. The Channel Division Method

In the design of channelized receivers, the prototype filter is not ideal, and there is a transition zone. According to whether the sub-channels overlap or not, the channel dividing methods can be divided into non-overlapping channel dividing and overlapping channel dividing, as shown in Figure 3. The non-overlapping channel dividing will form a receiving blind zone. When the signal is in the transition zone, the signal cannot be effectively received. Moreover, the cross-channel broadband signal may be misjudged as multiple narrowband signals due to the signal in the transition zone being lost. The overlapping channel division method overlaps different channels, which solves the problem of receiving blind spots. However, under critical sampling conditions, the channel sampling rate is $2\pi/K$. When overlapping channel dividing is used, the filter coverage bandwidth is greater than the channel sampling rate. The signal spectrum within the channel is aliased. The formula is derived as follows.

In discrete time, the instantaneous frequency can be approximated by the backward difference operation. The instantaneous frequency is

$$\hat{\omega}_k^i[m] = \frac{\Delta\phi_k[m]}{D} \tag{1}$$

where the $\wedge$ denotes the estimate. In order to have an unambiguous frequency measurement

$$-\pi \leq \Delta\phi_k[m] \leq \pi \tag{2}$$

Thus

$$-\pi \leq \hat{\omega}_k^i[m]D \leq \pi \tag{3}$$

Suppose the channel overlap factor is $\delta$. The frequency range of the signal in the channel can be expressed as

$$-\frac{\pi}{K} \cdot (1+2\delta) \leq \omega_k^i[m] \leq \frac{\pi}{K} \cdot (1+2\delta) \tag{4}$$

Using Equations (3) and (4), the IFM for each channel will be unambiguous if

$$F = K/D \geq (1+2\delta) \tag{5}$$

In addition, wideband signals can suffer from cross-channel problems [11].

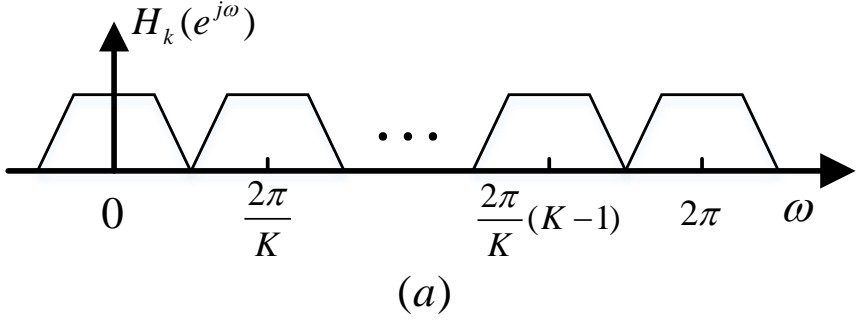

$(a)$

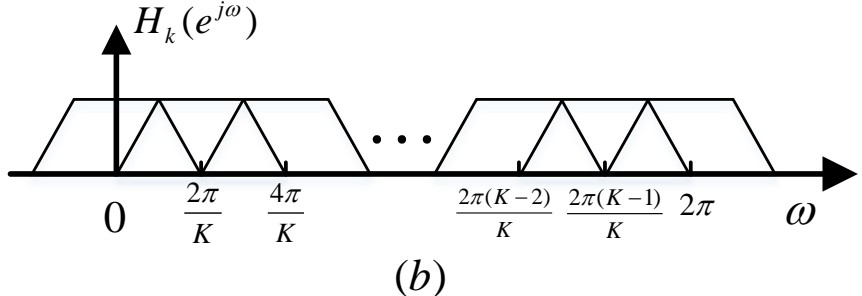

$(b)$

**Figure 3.** The channel division method: (**a**) non-overlapping channel dividing; (**b**) overlapping channel dividing.

### 2.2. Proposed Algorithm Description

This section proposes a digital channelized receiver architecture for wideband low SNR chirp signal detection. A non-critical sampling channelization structure is adopted, eliminating signal spectral aliasing in the channel. An adaptive threshold generation algorithm is proposed to detect signals without prior conditions. A CORDIC-based IFM algorithm is proposed, improving chirp signals' frequency estimation accuracy. Moreover, a channel arbitration logic is proposed to complete the cross-channel combination of wideband signals. The detailed flow of the proposed digital channelized receiver architecture is shown in Figure 4.

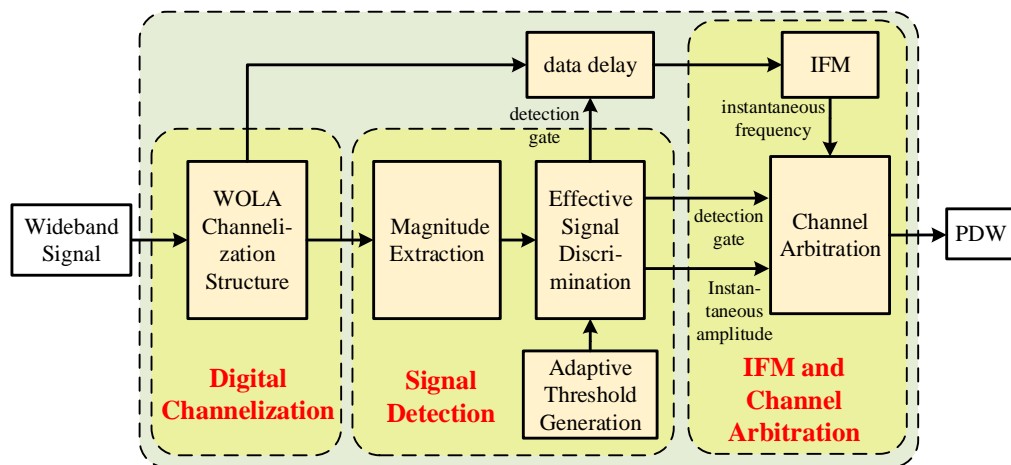

**Figure 4.** The detailed flow of the proposed digital channelized receiver architecture.

2.2.1. WOLA Channelization Structure

To develop the WOLA channelization structure, we begin with the low-pass channelization structure shown in Figure 2. Recall that the output signal $y_k(m)$ for the $k_{th}$ channel of the filter bank analyzer can be expressed in the form [33]

$$
\begin{aligned}
y_k(m) &= \sum_{n=-\infty}^{\infty} h(mD - n)x(n)exp(-j\omega_k n) \\
&= \sum_{n=-\infty}^{\infty} h(mD - n)x(n)W_K^{-kn}
\end{aligned}
\tag{6}
$$

where $\omega_k = 2\pi k/K$, $W_K = exp(j2\pi/K)$, $k = 0, 1, \cdots, K-1$.

According to the discrete short-time Fourier transform (STFT) definition, $y_k(m)$ can be regarded as the short-time spectrum at time $n = mD$. The filter $h(n)$ acts as a sliding analysis window that selects and weights the short-time segment of the signal $x(n)$ to be analyzed. In a practical implementation, the data slide by in time, and the filter is invariant. Thus, it is convenient to convert the data reference point from a fixed time frame to a sliding time frame, which is accomplished by the change of variables

$$
r = n - mD
\tag{7}
$$

Then, the short-time transform $y_k(m)$ can be expressed as [33]

$$
\begin{aligned}
y_k(m) &= \sum_{r=-\infty}^{\infty} h(-r)x(r+mD)W_K^{-k(r+mD)} \\
&= W_K^{-kmD} \sum_{n=-\infty}^{\infty} h(-r)x(r+mD)W_K^{-kr} \\
&= W_K^{-kmD} \tilde{X}_k(m)
\end{aligned}
\tag{8}
$$

where $\tilde{X}_k(m)$ is defined as STFT, that is

$$
\tilde{X}_k(m) = \sum_{n=-\infty}^{\infty} h(-r)x(r+mD)W_K^{-kr}
\tag{9}
$$

Let

$$
y_m(r) = h(-r)x(r+mD)
\tag{10}
$$

For efficient implementation of Equation (9) with the aid of the FFT algorithm, the sequence $y_m(r)$ should be time aliased into the form of a $K$-point sequence $x_m(r)$. The sequence $x_m(r)$ can be expressed as [33]

$$x_m(r) = \sum_{l=-\infty}^{\infty} y_m(r + lK) \tag{11}$$

So that Equation 9 becomes [33]

$$\tilde{X}_k(m) = \sum_{r=0}^{K-1} x_m(r) W_K^{-kr} \tag{12}$$

Thus

$$y_k(m) = W_K^{-kmD} \sum_{r=0}^{K-1} x_m(r) W_K^{-kr} \tag{13}$$

The WOLA channelization structure is obtained from the above process, as shown in Figure 5. The WOLA channelization structure implements the analysis window using a direct extraction structure, which improves the efficiency by $D$ times. In addition, $K$ filter channels share the same windowing process, increasing efficiency by a factor of $K$. Moreover, the FFT is efficient. The WOLA structure has no restrictions on the number of channels $K$ and the decimation multiple $D$. Thus, the decimation multiple can be flexibly adjusted according to the frequency response of the prototype filter to ensure accurate frequency measurement with the lowest channel sampling rate.

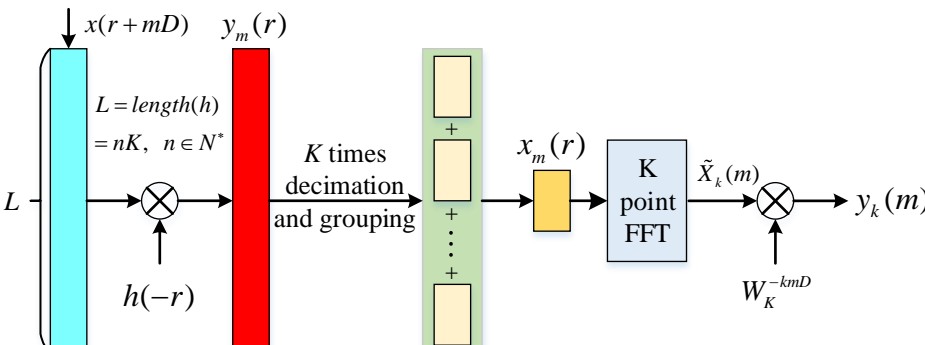

**Figure 5.** The weighted overlap-add (WOLA) channelization structure.

2.2.2. Adaptive Threshold Generation Algorithm

This section proposes a threshold calculation method, which generates the threshold by a moving window. The threshold is dynamic and is determined only by the instantaneous magnitude of the channelized signals. The window size is set to $P$, and the channelized output signal can be expressed as

$$y_k(m) = \sum_{n=0}^{\infty} y_k(m + nP), m = 1, 2, \cdots, P \tag{14}$$

Set $y_k(m + nP)$ as a data group. The same detection threshold is adopted for the same data group of each channel, and the detection threshold of the $n$th data group is generated from the $(n-1)$th data group. The calculation formula of the noise threshold $A_{TH}$ is as follows

$$A_{TH} = a \cdot A_N + W_N \tag{15}$$

where $A_N$ indicates the estimated noise value, $W_N$ indicates the minimum detection threshold value for suppressing the noise, and $a$ is the proportionality factor. The optimum values of the $W_N$ and $a$ are experimentally determined for different hardware platforms.

Thead channels are selected to calculate the noise threshold, considering the relevance of adjacent channel noise. In addition, in order to eliminate the noise influence of the DC component and the high-frequency component, it is necessary to discard the odd marginal channels. Calculate a set of amplitudes for each sampling point, select the smallest amplitude as the noise accumulation point, and accumulate $P$ points as the noise estimation value. The influence of random noise on the threshold is reduced by accumulating $P$ data points.

### 2.2.3. IFM and Channel Arbitration

A functional block diagram of IFM and channel arbitration is shown in Figure 6. The magnitude $A_k[m]$ and phase $\phi_k[m]$ of the channel output signal are extracted by a coordinate rotation digital computer (CORDIC) algorithm. The output signal after the channelized structure is the complex signal. Define $y_k[m] = I_k[m] + jQ_k[m]$; the instantaneous magnitude and phase are shown below [25].

$$A_k[m] = \sqrt{I_k^2[m] + Q_k^2[m]} \tag{16}$$

$$\phi_k[m] = \arctan \frac{I_k[m]}{Q_k[m]} \tag{17}$$

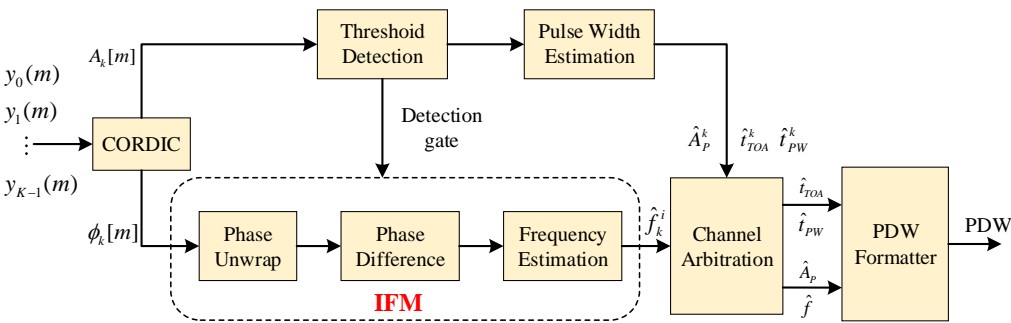

**Figure 6.** Channel detection, arbitration, and parameter estimation block diagram.

The instantaneous magnitudes are routed to the signal detection module and the instantaneous phases to the IFM module. The detection module provides a detection gate to the IFM module, which determines the channel where the valid signal is located. Considering the influence of the "rabbit-ear-effect", it is determined that there is a valid signal only when $Q$ consecutive sampling points exceed the detection threshold. The minimum pulse width, sampling rate, and channel number determine the value of $Q$. The phase value obtained by the CORDIC algorithm is $[-\pi, \pi]$. As the number of sample points increases, the actual phase of the signal will span periods, resulting in phase ambiguity. Therefore, the IFM needs to unwrap the phase before the frequency calculation.

Let $C_m$ be the phase compensation parameter at time $m$ and initialize $C_1 = 0$. The formula for phase deblurring is as follows [25]:

$$C_m = \begin{cases} C_{m-1} - 2\pi, & \phi_k[m] - \phi_k[m-1] \geq \pi \\ C_{m-1} + 2\pi, & \phi_k[m] - \phi_k[m-1] \leq -\pi \\ C_{m-1}, & \text{otherwise} \end{cases} \tag{18}$$

$$\phi'_k[m] = \phi_k[m] + C_m \tag{19}$$

The frequency output can be expressed as

$$\hat{f}^i_k[m] = \frac{f_{ch}}{2\pi}\Delta\phi'_k[m] = \frac{f_s}{2\pi \cdot D}\Delta\phi'_k[m] \tag{20}$$

The IFM error is significant with low SNR signals, which cannot meet the requirements. This study proposes a local flat mean algorithm, which eliminates frequency outliers by finding instantaneous frequency points that satisfy the frequency flat condition. The judgment condition of frequency flatness is that the adjacent frequency differences of 5 sampling points are all within the error range $\Delta f$, that is

$$\begin{cases} \left| \hat{f}^i_k[m+1] - \hat{f}^i_k[m] \right| < \Delta f \\ \left| \hat{f}^i_k[m+2] - \hat{f}^i_k[m+1] \right| < \Delta f \\ \left| \hat{f}^i_k[m+3] - \hat{f}^i_k[m+2] \right| < \Delta f \\ \left| \hat{f}^i_k[m+4] - \hat{f}^i_k[m+3] \right| < \Delta f \end{cases} \tag{21}$$

The setting of $\Delta f$ is related to the frequency interval between two adjacent sampling points. If the frequency modulation slope is $\gamma$ for a chirp signal, the frequency interval between two adjacent sampling points is $f_{step} = \gamma/f_{ch}$. The value of $\Delta f$ should be greater than $f_{step}$. In addition, the value of $\Delta f$ is related to the input SNR. When the SNR is low, the measured instantaneous frequency is more uneven, and the value of $\Delta f$ should be increased at this time [32]. The value of $\Delta f$ under different SNRs can be determined by experiment. A flat frequency vector $\vec{f}^{avg}_k$ can be generated by taking the local average of the frequency points satisfying the frequency flat condition. The signal bandwidth and center frequency are calculated by finding the maximum and minimum values of the flat frequency vector.

The pulse frequency estimates $\vec{f}^{avg}_k$ are then sent to the channel arbitration logic which determines whether adjacent channel signals belong to the same signal. The channel arbitration logic is shown in Equation (22), where $\Delta B$ is determined by the prototype filter response.

$$a = \left\{ \max(\vec{f}^{avg}_{k-1}) > \min(\vec{f}^{avg}_k) \right\} || \left\{ \min(\vec{f}^{avg}_k) - \max(\vec{f}^{avg}_{k-1}) < \Delta B \right\} \tag{22}$$

The signals in channel $(k-1)$ and channel $k$ are the same signal if

$$a = 1 \tag{23}$$

A different result for

$$a = 0 \tag{24}$$

After channel arbitration, the pulse parameters (frequency, amplitude, pulse width, and time-of-arrival) are fed into the pulse descriptor word (PDW) formatter and output.

## 3. Results and Discussion

This section verifies the performance of the proposed receiver architecture. In Section 3.1, the simulation experiments are conducted using MATLAB, including the filter bank structure, the channel arbitration logic, and the frequency estimation algorithm [32]. The performance of the IFM algorithm is also evaluated in Section 3.2. Section 3.3 gives the computational complexity analysis. Finally, a discussion is shown in Section 3.4.

### 3.1. Simulation Experiments

The system parameters and prototype filter parameters are shown in Table 2. The overlapping channel division method is adopted, and the channel overlap factor is 0.3. The filter structure is a non-critical sampling structure based on WOLA. The oversampling factor is 1.6, determined by the channel overlap factor.

Figure 7 shows the realized magnitude response for filter banks. The number of output channels is $K/2$. As shown in Figure 7, it is clear that there is no blind zone for the signal interception in the receive bandwidth. In addition, oversampling ensures no aliasing of the signal spectrum within the channel.

**Table 2.** System parameters and prototype filter parameters.

|  | Parameter Type | Parameter Value |
|---|---|---|
| System parameters | Sampling Rate ($f_s$) | 1200 MHz |
|  | Channel number | 32 |
|  | Decimation number | 20 |
| Filter parameters | Passband frequency | 19.5 MHz |
|  | Stopband frequency | 28.5 MHz |
|  | Filter order | 256 |

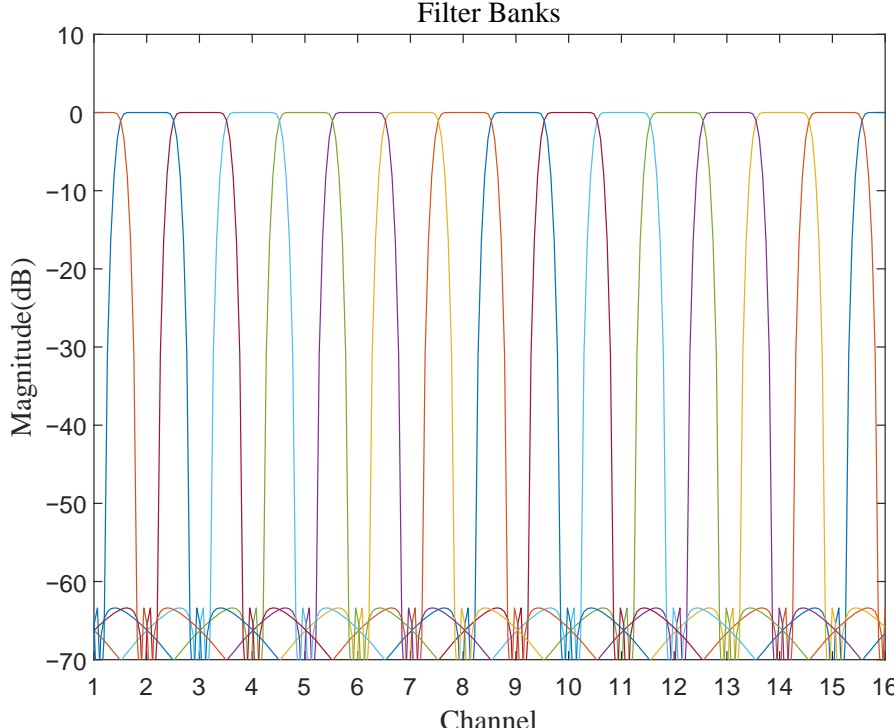

**Figure 7.** The magnitude frequency response of filter banks. (Different color lines indicate the frequency response of the bandpass filter for different channels).

For the purpose of demonstration, the input signal parameters are shown in Table 3. The input signal for simulation includes three chirp signals. The first two chirp signals fall within the channel, and the third is distributed across the channel. The SNR for all three signals is 0 dB. The signal processing is shown below.

**Table 3.** Input signal parameters.

| Parameter Type | Signal 1 | Signal 2 | Signal 3 |
|---|---|---|---|
| Modulation type | Chirp | Chirp | Chirp |
| SNR | 0 dB | 0 dB | 0 dB |
| Bandwidth | 20 MHz | 20 MHz | 100 MHz |
| Center frequency | 40 MHz | 70 MHz | 185 MHz |
| Pulse width | 5 μs | 5 μs | 20 μs |

The WOLA channelization results are shown in Figure 8. For convenience, only the amplitude response of the first 16 channels is shown. Based on the adaptive detection threshold and the instantaneous magnitude, the correct channels are identified as 2, 3, 5, 6, and 7. The IFM of the correct channels are shown in Figures 9 and 10. Figures 9 and 10 show that the original IFM is sensitive to noise and has poor frequency estimation accuracy when the SNR is low. After processing by the local flat mean algorithm, the frequency estimation accuracy improves significantly. The frequency estimation results for the correct channels are as follows

$$
\begin{cases}
f_{BW2} = 19.95 \text{ MHz}, f_{CF2} = 2.81 \text{ MHz} \\
f_{BW3} = 19.68 \text{ MHz}, f_{CF3} = -5.08 \text{ MHz} \\
f_{BW5} = 35.74 \text{ MHz}, f_{CF5} = 3.27 \text{ MHz} \\
f_{BW6} = 41.64 \text{ MHz}, f_{CF6} = 0.12 \text{ MHz} \\
f_{BW7} = 33.06 \text{ MHz}, f_{CF7} = -6.27 \text{ MHz}
\end{cases}
\tag{25}
$$

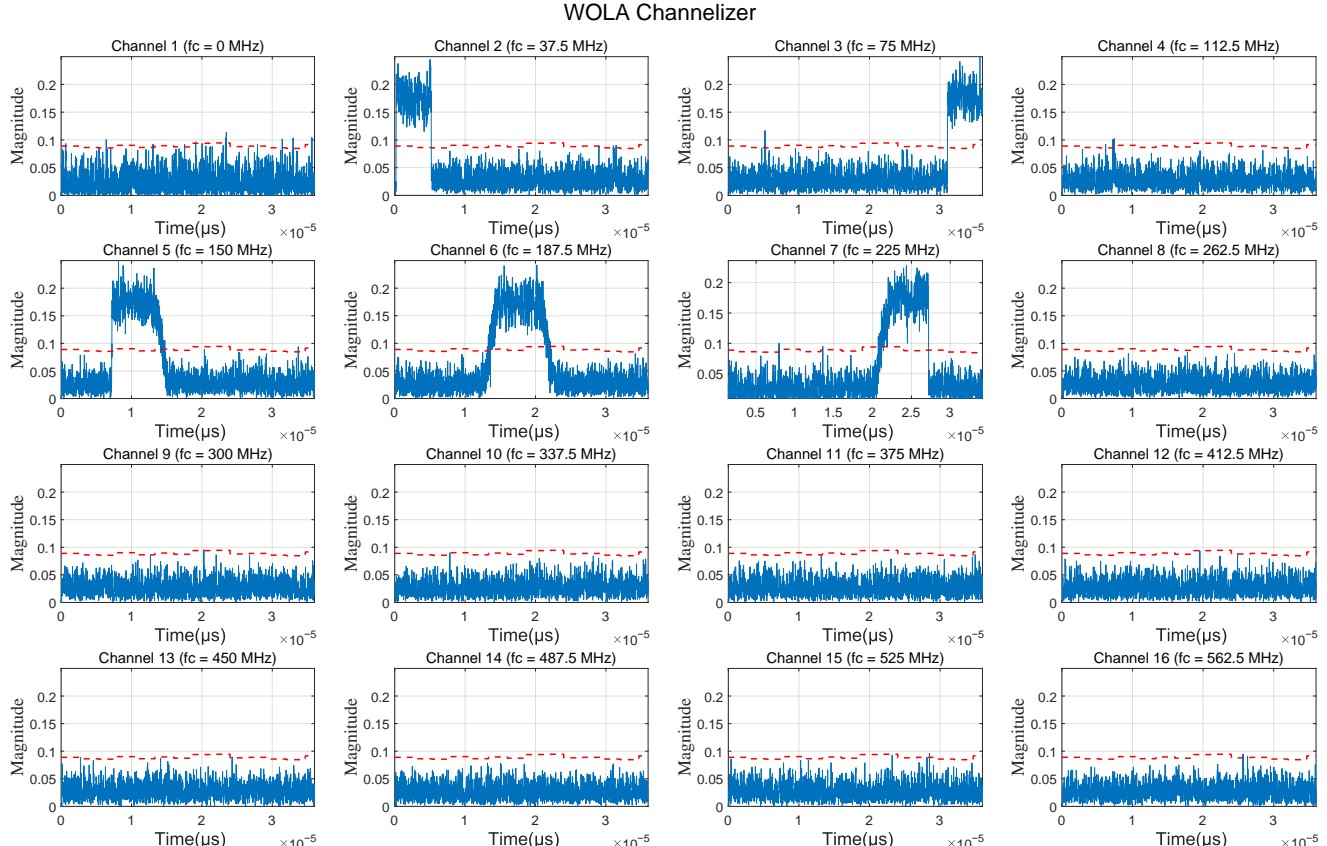

**Figure 8.** The WOLA channelization results. (The blue line indicates the instantaneous amplitude of the signal in the channel, and the red dashed line indicates the detection threshold).

Introduce the the channel centre frequency to obtain the start and end frequencies of the signal in the channel. The results can be expressed as

$$
\begin{cases}
f_{st2} = 30.34 \text{ MHz}, f_{sp2} = 50.29 \text{ MHz} \\
f_{st3} = 60.08 \text{ MHz}, f_{sp3} = 79.76 \text{ MHz} \\
f_{st5} = 135.40 \text{ MHz}, f_{sp5} = 171.14 \text{ MHz} \\
f_{st6} = 166.80 \text{ MHz}, f_{sp6} = 208.44 \text{ MHz} \\
f_{st7} = 202.20 \text{ MHz}, f_{sp7} = 235.26 \text{ MHz}
\end{cases}
\tag{26}
$$

Channel arbitration is then completed based on the IFM results. The results show that the intercepted mixed pulse stream consists of three separate signals. The signals within channels 5, 6, and 7 belong to the same signal. The estimation results of signal parameters are shown in Table 4. Simulations show that the proposed receiver architecture is reliable for low SNR chirp signal interception with high parameter estimation accuracy.

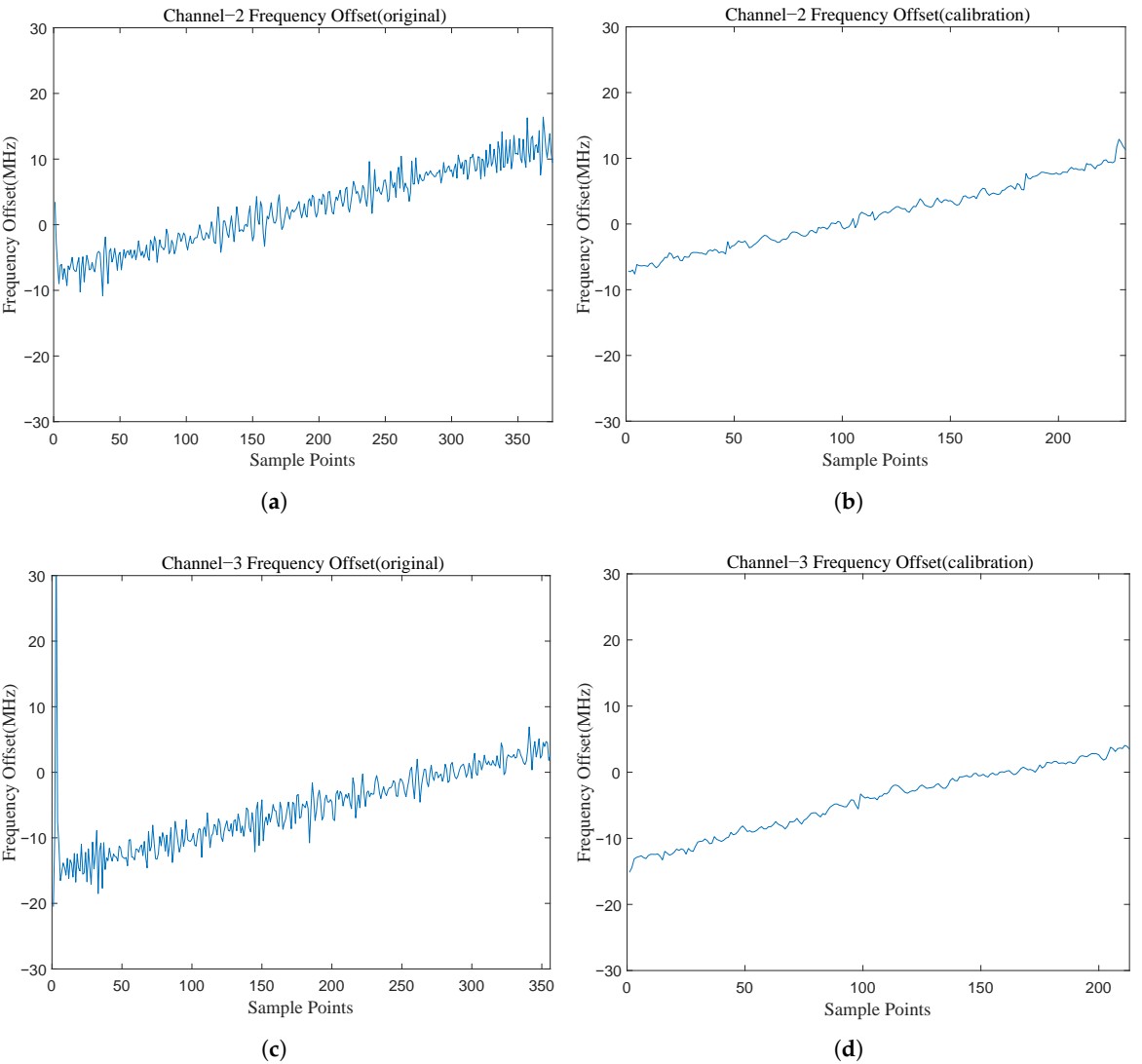

**Figure 9.** The instantaneous frequency measurement (IFM) results of the correct Channel 2 and 3: (**a**) the original IFM result of Channel 2; (**b**) the calibrated IFM result of Channel 2; (**c**) the original IFM result of Channel 3; (**d**) the calibrated IFM result of Channel 3.

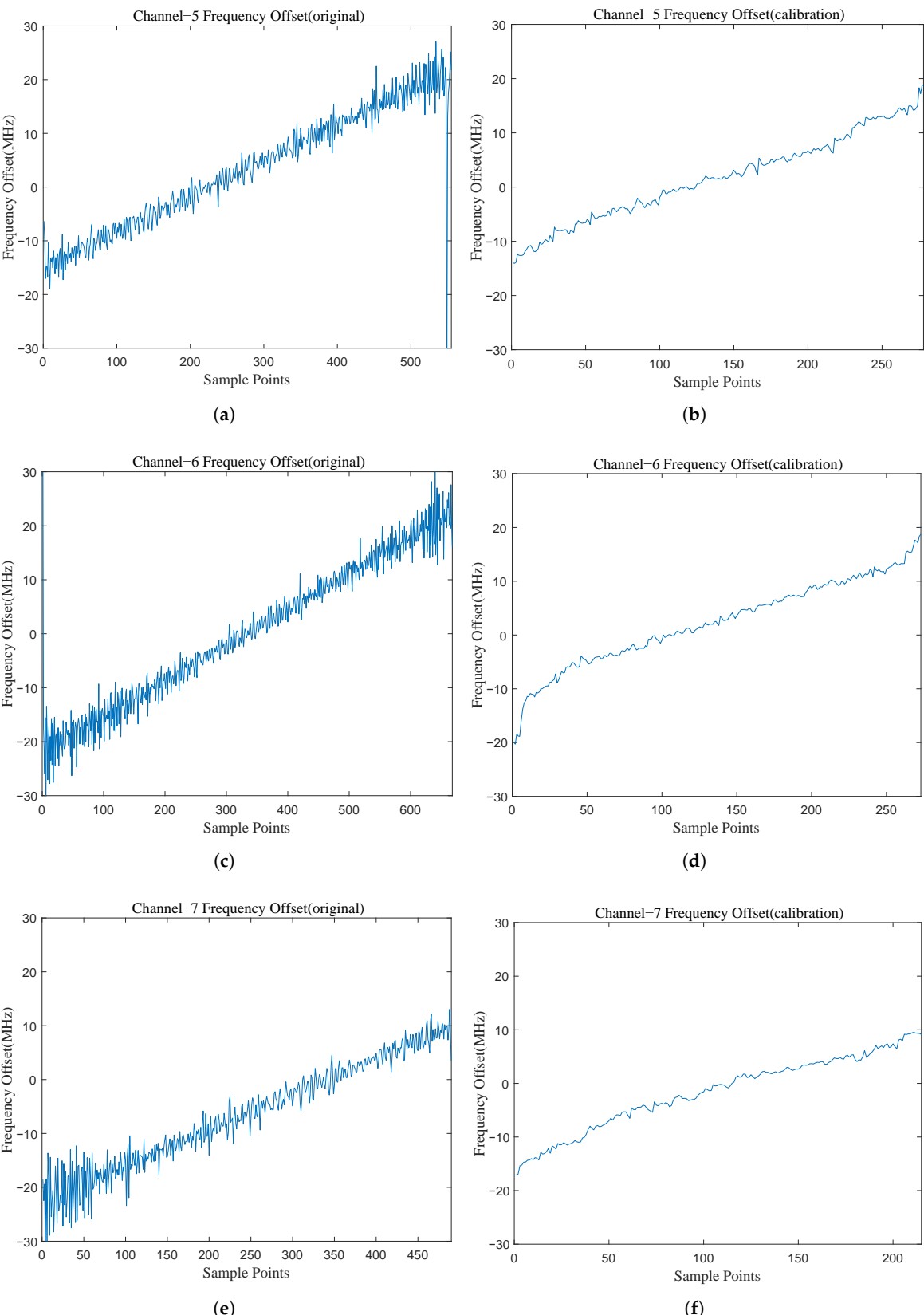

**Figure 10.** The IFM results of the correct Channel 5, 6, and 7: (**a**) the original IFM result of Channel 5; (**b**) the calibrated IFM result of Channel 5; (**c**) the original IFM result of Channel 6; (**d**) the calibrated IFM result of Channel 6; (**e**) the original IFM result of Channel 7; (**f**) the calibrated IFM result of Channel 7.

**Table 4.** The estimated results.

| Parameter Type | Signal 1 | Signal 2 | Signal 3 |
|---|---|---|---|
| Bandwidth | 19.95 MHz | 19.68 MHz | 99.86 MHz |
| Center frequency | 40.31 MHz | 69.92 MHz | 183.33 MHz |
| Pulse width | 5 µs | 5 µs | 20 µs |

In order to facilitate the display of simulation results, the bandwidth of the chirp signal used in the simulation is insignificant. The instantaneous bandwidth that the receiver architecture proposed in this paper can adapt to is much larger than that adopted by the simulation. The system sampling rate determines the instantaneous bandwidth. The higher the system sampling rate, the larger the instantaneous bandwidth the receiver can handle. The band-pass sampling theorem limits the specific relationship between the instantaneous bandwidth and the system sampling rate.

*3.2. Performance Evaluation of the Proposed IFM Algorithm*

This subsection analyzes the proposed IFM algorithm's performance under different input SNRs. Refs. [17,32] were used for comparison. Because the frequency of each sampling point changes, the average processing cannot be performed directly for the frequency estimation of wideband chirp signals. In this case, the results of the frequency estimation algorithm in Refs. [17,32] are the same. System parameters are set according to Table 2. Chirp signal bandwidth and center frequency are set to 100 MHz. The frequency root-mean-square error (RMSE) is calculated by the following formula

$$RMSE = \sqrt{\frac{1}{N}\sum_{i=1}^{N}(\hat{f}_i - f)^2} \tag{27}$$

As shown in Figure 11, the frequency estimation accuracy is excellent when the input SNR is higher than 0 dB. When the input SNR is below 0 dB, the performance of the proposed algorithm decreases because the noise interference is serious, even though the filter bank has improved the SNR. The improved IFM algorithm cannot thoroughly suppress noise interference under this condition. Compared with Refs. [17,32], the IFM algorithm proposed in this paper has higher estimation accuracy. The accuracy difference is not apparent when the input SNR is higher than 8 dB. The accuracy difference becomes significant when the SNR is lower than 8 dB. When the input SNR becomes higher, the algorithm based on instantaneous phase measurement is less affected by noise. When the input SNR is lower than 8 dB, the noise interference to the instantaneous phase becomes serious. However, the IFM algorithm proposed in Refs. [17,32] cannot suppress the noise interference when the input signal is a chirp signal, it can only maintain good estimation accuracy for point frequency signal when the SNR is low. The IFM algorithm proposed in the study can suppress noise interference by the local flat mean algorithm and improve the estimation accuracy.

In addition, the frequency RMSE of the bandwidth is greater than the center frequency. The reason is that the sampling points located at the rising and falling edges of the signal are more affected by noise. Under this condition, the maximum frequency measured by the algorithm becomes small, and the minimum frequency becomes large. This result significantly impacts the bandwidth measurement's accuracy more than the center frequency measurement. For that

$$\begin{cases} f_{BW} = f_{max} - f_{min} \\ f_{CF} = \frac{f_{max}+f_{min}}{2} \end{cases} \tag{28}$$

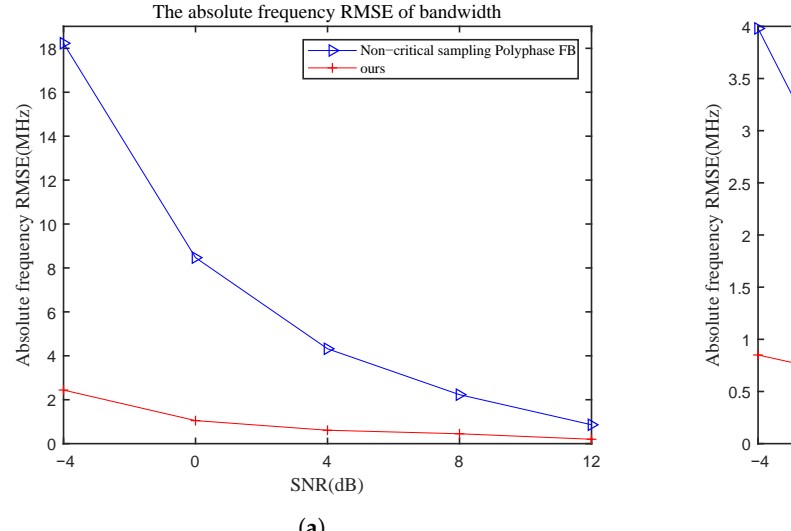
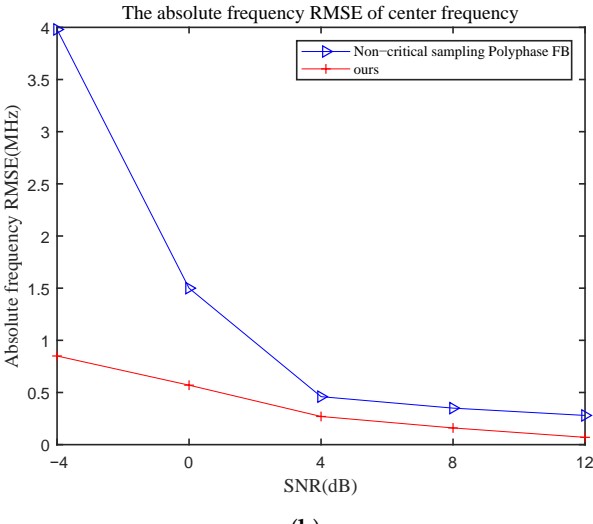

(**a**)                                         (**b**)

**Figure 11.** The absolute frequency root-mean-square error (RMSE) of bandwidth and center frequency with different SNR: (**a**) the absolute frequency RMSE of bandwidth; (**b**) the absolute frequency RMSE of center frequency. (The blue line indicates the results of the Refs. [17,32]; The red line indicates the results of the algorithm proposed in this paper).

### 3.3. Computational Costs

This subsection analyzes the computational costs of the WOLA channelization structure. The conventional digital down conversion (DDC) [9] and polyphase filter [13] structures are used for comparison. Suppose the number of sample points is $N$, the number of channels is $K$, the decimation factor is $D$, and the order of the prototype filter is $L$.

The three methods all adopt the critical sampling complex channel structure. The number of multipliers consumed by three channelized structures is shown in Table 5. As shown in Table 5, the channelized structure adopted in this paper is as efficient as the polyphase filter structure and superior to the DDC structure. In addition, the IFM algorithm adopted in this paper is efficient because the instantaneous frequency is calculated only by phase difference value.

**Table 5.** The computational complexity of the three channelized structures.

| Channelized Structures | Number of Multipliers |
|---|---|
| DDC structure [9] | $NK(L+1)$ |
| polyphase filter structure [13] | $N(2 + L/K + \log_2 K)$ |
| ours | $N(1 + L/K + \log_2 K)$ |

### 3.4. Discussion

This subsection summarizes the advantages of the proposed algorithm in this study compared to the existing works. In addition, the limitations of the proposed algorithm are presented.

The receiver architecture proposed in this study is computationally efficient and can accommodate a wide instantaneous bandwidth, just like the receiver architecture proposed in Refs. [17,32]. In addition, the receiver architecture proposed in this paper has three advantages over the Refs. [17,32]. First, the proposed receiver architecture uses an adaptive threshold detection algorithm that does not require a priori information for signal detection. Second, the channelization sample rate can precisely match the prototype filter bandwidth due to adopting the WOLA-based channelization structure. Third, the estimation accuracy for the chirp signal is higher when the SNR is lower than 8 dB.

However, there is a limitation to the proposed algorithm. As shown in Figure 11, when the input SNR is below 0 dB, the performance of the proposed algorithm decreases. The principle of the algorithm determines this. This paper calculates the instantaneous frequency through the instantaneous phase, and it is computationally efficient and easy to implement. However, the instantaneous phase is heavily affected by noise. Although the algorithm adopted in this study can suppress the influence of noise to a certain extent, the estimation accuracy of the algorithm is limited when the noise is large enough. The algorithm proposed in this paper cannot meet the requirements for high-precision detection of ultra-low SNR signals.

## 4. Conclusions and Recommendation

This paper proposes an efficient receiver architecture for implementing a digital channelized receiver for low SNR chirp signals intercept applications. By optimizing the factor $F$ in the relation $K = FD$, the channel arbitration logic is shown to be reliable for cross-channel signals. Based on the channel arbitration logic, the proposed receiver architecture can adapt to the instantaneous bandwidth limited only by the system sampling rate and the band-pass sampling theorem. A CORDIC-based instantaneous frequency measurement algorithm is also proposed. When the input SNR is 0 dB, the absolute frequency RMSE of bandwidth and the center frequency are 0.57 MHz and 1.05 MHz, respectively. In addition, an adaptive threshold generation algorithm is proposed to detect signals without prior information. Simulation shows that the proposed algorithm is reliable and robust for low SNR and wideband chirp signal detection.

The contributions of this paper are summarized as follows.

1. A non-critical sampling digital channelized receiver architecture is proposed to detect wideband low SNR chirp signals. The proposed receiver architecture can adapt to a wide instantaneous bandwidth with high frequency estimation accuracy when the SNR is greater than 0 dB.
2. An adaptive threshold generation algorithm is proposed to detect signals without prior information.
3. A CORDIC-based instantaneous frequency measurement algorithm is also proposed, improving low SNR chirp signals' frequency estimation accuracy.

The efficient frequency measure algorithms for lower SNR chirp signals in the future need further research. It is possible to combine instantaneous frequency measurement algorithms with existing high-precision parameter estimation algorithms to achieve high-precision estimation with high efficiency.

**Author Contributions:** Conceptualization, W.C. and Q.Z.; methodology, W.C. and W.L.; validation, W.L., H.W. and W.C.; software, H.W. and W.C.; writing—original draft preparation, W.C.; writing—review and editing, Q.Z.; visualization, W.C.; supervision, X.L.; project administration, Q.Z. and X.L.; funding acquisition, Q.Z. All authors have read and agreed to the published version of the manuscript.

**Funding:** This research received no external funding.

**Institutional Review Board Statement:** Not applicable.

**Informed Consent Statement:** Not applicable.

**Data Availability Statement:** Not applicable.

**Acknowledgments:** The authors would like to thank the editors and reviewers for their efforts to help the publication of this work.

**Conflicts of Interest:** The authors declare no conflict of interest.

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
