# Peer review of "An Efficient Digital Channelized Receiver for Low SNR and Wideband Chirp Signals Detection"

_applsci, doi:10.3390/app13053080_

Round 1
Reviewer 1 Report
The topic is interesting and worth to be analysed. In general, the manuscript has a good structure. However, to be accepted as a research paper, the following shortcomings should be addressed. First, the abstract section should be revised because it is not very clear what the main outcomes of this study are. Additionally, it is not clear what is the novelty of the review paper in comparison to what already present in the specific literature. This is very important. Even if the content of the subsections is quite well presented, the subsections about strategies should be further extended also in terms of number of reference, if possible, and so as requested by review papers. The conclusion sections should have the structure of bullet points summarizing the main outcomes of this study.
1. The abstract is not detailed enough. Readers expect to see more detail of the methodology, results, and conclusion in the abstract. The abstract need to be greatly improved. The abstract does not show that the authors achieved much as there is no numerical justification to back the author’s claims or results of comparative analysis to show superior performance.
2. In the introduction, the authors should explain why they did it (motivation) discussing the possible outcome. Readers are primarily interested in the motivation and outcome of your research. Therefore, a good introduction should contain:
a. What is the problem to be solved?
b. Are there any existing solutions?
c. Which is the best?
d. What is the main limitation of the best and existing approaches?
e. What do you hope to change or propose to make it better?
f. How is the paper structured?
3. Please clearly highlight how your work advances the field from the present state of knowledge and you should provide a clear justification for your work which should be stated at the end of literature review/ related works. The impact or advancement of the work can also appear in the conclusion.
4. Related works section is not sufficient. The authors should improve on this section as they have left many papers out. Normally, it’s the gaps in work of others that the authors are expected to fill. Therefore, at the end of your review section state the problems in this field with appropriate reference and tell readers which one your work addresses.
The authors should consult and cite:
- Interference mitigation technique for self-optimizing Picocell indoor LTE-A networks. Telecommunication Systems, vol. 81, 549-560, 2022. DOI: 10.1007/s11235-022-00966-3
- An efficient wide-band signal detection and extraction method, 2021, https://www.matec-conferences.org/articles/matecconf/pdf/2021/05/matecconf_cscns20_04011.pdf
- Genetic Algorithm Based Optimum Finger Selection for Adaptive Minimum Mean Square Error Rake Receivers Discrete Sequence-CDMA Ultra-Wide Band Systems. Wireless Personal Communication, vol. 123, pp. 1537–1551, 2022. DOI: 10.1007/s11277-021-09199-0
5. There is no comparison of results with the existing works in this paper. This should be added for readers to see how your proposed method performs relative to other works.
6. The authors should structure the paper into abstract, introduction, literature review/related works, methodology, results and discussion, and conclusion.
7. I was hoping to see more results and discussion as more results could be presented to make the work much appreciable. The authors are encouraged to reduce the plagiarism of the paper.
8. The Limitations of the proposed study need to be discussed before conclusion.
9. Some of the challenges encountered during the course of the study can be highlighted and future recommendations can be added at the end of the conclusion. Retitle conclusion as conclusion and recommendation.
10. The results and discussion section is weak. The authors should endeavor to improve on this section. In the section of selection of local minima, what criteria did the authors used. Also what priors did the authors consider? What is the minimum and the maximum values? If these are suitable, do they work for different types of images or just the images under consideration?
Author Response
We are very grateful for your review and professional comments on our article. Your comments will help improve the quality of the manuscript and have important guiding significance for our further research. According to your suggestions, we have made extensive corrections to our previous manuscript. The answers to each specific point are listed in response letter.
Reviewer 2 Report
The paper is new and technically sounds. Results somehow does support the methodology. The paper is properly organized, good literature review, suitable motivation and clear explanation on results are positive points to that. I think the abstract needs to be rephrased after revision to add some comments about any artifacts or negative points in the method, if exist. The introduction section is motivating, but needs to be improved. The methodology, results, and conclusions are well and completely developed, but needs to be minorly modified and developed according to the technical comments. Please also indicate if the codes available for this research? As I found, there is no code available for this study, e. g. in Github. If the authors could make the codes available, the manuscript could be much better evaluated, not only for reviewers, but also for possible readers. When it is not possible to upload the code for public access, such as in Github, could they be provided for reviewer for better assessment of the study? The theoretical background has been well explained in details, and the experiments and related models are presented. The result comparison parts are well organized and presented.
Author Response
We are very grateful for your review and professional comments on our article. Your comments will help improve the quality of the manuscript and have important guiding significance for our further research. According to your suggestions, we have made corrections to our previous manuscript. The answers to each specific point are listed in response letter.

Reviewer 3 Report
1-Title: differs from "A Digital Channelized Receiver for Low SNR and Wideband Chirp Signals Detection" from the pdf file, what is the correct?
2-The authors should make a table for the symbols to be clarified to the readers.
3-You can use these references as a support for the idea:
-On the hydrodynamic interaction of two coaxial spheres oscillating in a viscous fluid with a slip regime,ZAMM - Journal of Applied Mathematics and Mechanics, 1-16
-The axisymmetric migration of an aerosol particle embedded in a Brinkmann medium of a couple stress fluid with slip regime, European Journal of Pure and Applied Mathematics 15 (4), 1566–1592.
-The force on a magneto-spherical particle oscillating in a viscous fluid perpendicular to an impermeable planar wall with slippage,EUROPEAN JOURNAL OF PURE AND APPLIED MATHEMATICS 15 (3), 1376-1401
Author Response

(The authors gave the same response as above.)

Reviewer 4 Report
The manuscript:
“An Efficient Digital Channelized Receiver for Low SNR and Wideband Chirp Signals Detection”, by W. Cheng, Q. Zhang, W. Lu, H. Wang and X. Liu (Ref. No.: applsci-2203311-peer-review-v1),
contains interesting material. However, it is not well-organized and should be greatly elaborated. In particular, the authors claimed only advantages in their proposed method and simulation. However, they did not discuss about drawbacks that may appear, especially at higher SNR conditions. Therefore, it would be very desirable to include this discussion as well in order to estimate the efficiency of the proposed method/algorithm.
English is acceptable. However, the manuscript requires some citations.
Apart from this, the following should be taken into consideration:
Abstract
1) The key quantitative results obtained in this study should be reflected in the Abstract.
1. Introduction
1) The sentence: “In the published literature [13–15], the IFM module is mainly implemented by the CORDIC algorithm, which is simple and efficient”. The abbreviation for the CORDIC should be defined.
2) The sentence: “Nevertheless, the frequency measurement error can not meet …”. The verb “cannot” should be written without space.
3) The sentence: “Moreover, a Coordinate Rotation Digital Computer (CORDIC) based instantaneous frequency measurement (IFM) algorithm is proposed …”. CORDIC should be defined earlier.
2. Problem Formulation
1) The sentence: “The channelization method uses a digital filter bank to convert the sampled high-speed data into a baseband signal and decimate it simultaneously, reducing the data rate so that the signal can be real-time analyzed and processed by the signal processor”, should be cited.
2) The sentence: “Moreover, the cross-channel broadband signal will be misjudged as multiple narrowband signals”, is not clear and needs clarification.
3) The sentence: “However, traditional digital channelization structures use critical sampling for efficient implementation”, is also unclear and needs some clarification.
4) The sentence: “addition, wideband signals can suffer from cross-channel problems”, should be cited.
3. Proposed Algorithm Description
1) The major equations should be cited.
2) The sentence: “The window size is set to P, and the channelized output signal can be expressed as …”. It is not clear why the channelized output signal can be expressed in form of equation (13).
4. Simulation and Analysis
1) The sentence: “In summary, the method proposed in this paper has better estimation accuracy under lower SNR conditions compared with [29]”. What is “under lower SNR conditions”. The specific range for lower SNR conditions should be provided.
2) What about higher SNR conditions? Does it provide better results at higher SNR conditions as compared to [29]?
3) The meaning of the sentence: “The complex algorithm is also not appropriate for the application of this study”, is not clear.
5. Conclusions
1) Similar to Abstract, the section Conclusions should also provide the key quantitative results, obtained in this study.
The manuscript requires a major mandatory revision.
Author Response

(The authors gave the same response as above.)

Round 2
Reviewer 1 Report
The authors should endeavour to add a comparative analysis with existing works and proofread the paper once again.
Author Response
We are very grateful for your review and professional comments on our article. Your comments will help improve the quality of the manuscript and have important guiding significance for our further research. According to your suggestions, we have made corrections to our previous manuscript.

Reviewer 4 Report
The manuscript:
“An Efficient Digital Channelized Receiver for Low SNR and Wideband Chirp Signals Detection”, W. Cheng, Q. Zhang, W. Lu, H. Wang and X. Liu (Ref. No.: applsci-2203311_v2),
has been improved after revision. However, a few minor corrections are still needed. In particular, the following should be taken into consideration:
1) The sentence: “By reducing the signal extraction multiple, the channel sampling rate is increased to ensure that the channel sampling rate is greater than the filter bandwidth”, should be cited.
2) The sentence: “The dynamic channelization method requires one more step of wideband signal reconstruction than the conventional method, which consumes many computing resources to meet the complete reconstruction conditions”, is confusing and requires a clarification. In particular, it is not clear why the dynamic channelization method needs an additional step of the wideband signal reconstruction.
3) The sentence: “The signal parameters are merged through IFM results” is not clear and requires clarification.
4) Table 1 requires more description.
5) The main equations in the subsection ‘2.2.1. WOLA Channelization Structure’ should be cited.
6) The sentence: “When the SNR is low, the measured instantaneous frequency is more uneven, and the value of Df should be increased at this time”, needs a citation.
7) The sentence: “This section verified the performance of the proposed receiver architecture. In subsection 4.1, the simulation experiments were done using MATLAB, including the filter bank structure, the channel arbitration logic, and the frequency estimation algorithm”, requires a citation.
8) The sentence: “In order to display the simulation results, the bandwidth of the chirp signal used in 280
the simulation is insignificant”. Why it is insignificant?
9) The subsection ‘3.4. Discussion’ is too short and not informative. It should be extended.
The manuscript requires a minor revision.
Author Response

(The authors gave the same response as above.)
